# Mapping Local Variations and the Determinants of Childhood Stunting in Nigeria

**DOI:** 10.3390/ijerph20043250

**Published:** 2023-02-13

**Authors:** Kedir Y. Ahmed, Allen G. Ross, Seada M. Hussien, Kingsley E. Agho, Bolajoko O. Olusanya, Felix Akpojene Ogbo

**Affiliations:** 1Rural Health Research Institute, Charles Sturt University, Orange, NSW 2800, Australia; 2Translational Health Research Institute, Western Sydney University, Campbelltown, Locked Bag 1797, Penrith, NSW 2751, Australia; 3School of Public Health, College of Medicine and Health Sciences, Wollo University, Dessie 1145, Ethiopia; 4School of Health Sciences, Western Sydney University, Campbelltown Campus, Locked Bag 1797, Penrith, NSW 2751, Australia; 5Centre for Healthy Start Initiative, 286A Corporation Drive, Dolphin Estate, Ikoyi, Lagos 101223, Nigeria; 6Riverland Academy of Clinical Excellence (RACE), Riverland Mallee Coorong Local Health Network, SA Health|Government of South Australia, Berri, SA 5343, Australia

**Keywords:** children, stunting, malnutrition, geospatial distribution, Nigeria

## Abstract

Introduction: Understanding the specific geospatial variations in childhood stunting is essential for aligning appropriate health services to where new and/or additional nutritional interventions are required to achieve the Sustainable Development Goals (SDGs) and national targets. Objectives: We described local variations in the prevalence of childhood stunting at the second administrative level and its determinants in Nigeria after accounting for the influence of geospatial dependencies. Methods: This study used the 2018 national Nigeria Demographic and Health Survey datasets (NDHS; N = 12,627). We used a Bayesian geostatistical modelling approach to investigate the prevalence of stunting at the second administrative level and its proximal and contextual determinants among children under five years of age in Nigeria. Results: In 2018, the overall prevalence of childhood stunting in Nigeria was 41.5% (95% credible interval (CrI) from 26.4% to 55.7%). There were striking variations in the prevalence of stunting that ranged from 2.0% in Shomolu in Lagos State, Southern Nigeria to 66.4% in Biriniwa in Jigawa State, Northern Nigeria. Factors positively associated with stunting included being perceived as small at the time of birth and experience of three or more episodes of diarrhoea in the two weeks before the survey. Children whose mothers received a formal education and/or were overweight or obese were less likely to be stunted compared to their counterparts. Children who were from rich households, resided in households with improved cooking fuel, resided in urban centres, and lived in medium-rainfall geographic locations were also less likely to be stunted. Conclusion: The study findings showed wide variations in childhood stunting in Nigeria, suggesting the need for a realignment of health services to the poorest regions of Northern Nigeria.

## 1. Introduction

Childhood malnutrition is a major global health concern that affects the growth and development of children and has long-lasting impacts into adulthood [1,2]. The causes of malnutrition are multi-faceted and interconnected, and include poverty, poor hygiene, limited access to nutritious food and healthcare, and infectious diseases [2,3,4,5]. Addressing malnutrition is important for improving health outcomes and breaking the cycle of poverty worldwide.

The United Nations General Assembly endorsed the Sustainable Development Goals (SDGs, including SDG-2.2: to end all forms of malnutrition by 2030) in 2015 [6]. The World Health Organisation member states endorsed the Global Nutrition Targets (GNTs; Target 1: to reduce the number of children under 5 who are stunted by 40% before 2025) in 2012 [7]. These efforts aim to identify priority areas for action and serve as catalysts for global change [6,8]. However, current reports indicate that an estimated 144 million children under 5 are stunted globally [9,10], and no African country (including Nigeria) is on track to achieve the SDGs and GNTs [10].

Nigeria is a multi-ethnic and culturally diverse country in the sub-Saharan region of West Africa, with an estimated population of over 210 million people [11]. In the past two decades, the country has made steady economic progress, with an increase in Growth Domestic Product (GDP) per capita income from USD 568 in 2000 to USD 2085 in 2021 [12]. Despite this economic growth, more than 40% of Nigerians still live in poverty, and childhood malnutrition (that is, stunting, wasting, and underweight) remains a huge public health challenge [13]. Recent reports have indicated that childhood malnutrition accounts for one out of three cases of mortality of children aged under 5 in Nigeria [13,14].

For the past two decades, there have been global, national, and subnational efforts to improve childhood malnutrition in Nigeria. Some of the programmes include the Millennium Development Goals [15], Nigeria’s National Policy on Food and Nutrition (MBNP, 2016) [16], and the Child Development Grant Programme in Northern Nigeria [17]. Many socioeconomic, agricultural, and health programmes have contributed to reducing the prevalence of childhood stunting in Nigeria (from 42.4% in 2003 to 36.8% in 2018) [13]. However, in 2021, the United Nations Children’s Fund (UNICEF) reported that approximately 12 million children were stunted in Nigeria, and the country accounted for 8% of the world’s stunted children [18].

Previously published studies conducted in Nigeria have shown that low maternal involvement in household decision making [19], a lack of parental formal education [20,21], child gender (male) [20,21], perceived birth size (small birth size) [22], low maternal body mass index (BMI, <18.5 kg/m^2^) [21], and household wealth (poorer households) [19,20,22] were associated with childhood stunting. Other significant factors included childhood diarrhoea [20,22], less-frequent antenatal visits [21], short breastfeeding duration (≤12 months) [19,22], and overcrowding [21]. While the previous studies provide a relevant description of the factors associated with stunting in Nigeria, the studies have limitations. Firstly, childhood malnutrition (including stunting) is often described as a geospatial public health issue [23]. In Nigeria, geography-specific variations in stunting have not been fully described, nor has there been an exploration of environmental and climatic factors (including temperature, rainfall, aridity, and urbanisation) associated with stunting in Nigeria. Secondly, none of the previous studies described detailed subnational estimates of stunting in Nigeria, as national data can often mask within-country geographic distribution [24]. Finally, understanding the geospatial variations in stunting can be the catalyst required to draw the attention of policy decision makers and public health experts to this chronic issue of stunting in Nigeria [25]. This measure can also increase efforts that aim to scale-up nutrition-specific interventions for Nigerian children and subsequently reduce malnutrition.

In 2020, the Federal Government of Nigeria launched the revised National Health Promotion Policy (NHP) [26], and it aims to deliver healthcare that is preventive, promotive, protective, restorative, and rehabilitative for every citizen, including improvement in childhood malnutrition. Findings from the current study would help determine high-priority geographic locations for targeted interventions needed to address Nigeria’s childhood stunting issue [26]. Furthermore, this study will be important in tracking progress at the subnational level in achieving the global nutritional agenda, including the SDGs [6] and GNTs [7] in Nigeria. The present study aims to (i) describe the prevalence of childhood stunting at the second administrative level in Nigeria, and (ii) to examine proximal and contextual factors associated with childhood stunting after accounting for the influence of geospatial dependencies.

## 2. Materials and Methods

### 2.1. Data Sources

The study was based on nationally representative data, the 2018 Nigeria Demographic and Health Survey (NDHS, N = 12,627). The National Population Commission (NPC) and the National Malaria Elimination Programme (NMEP) of the Federal Ministry of Health, Nigeria, implemented the survey. The 2018 NDHS was funded by the United States Agency for International Development (USAID), the Global Fund, Bill and Melinda Gates Foundation (BMGF), the United Nations Population Fund (UNFPA), and the World Health Organisation (WHO), with technical assistance from the Inner City Fund (ICF) International [13].

The Nigerian government structure is divided into 36 states and the Federal Capital Territory, Abuja. Each state is subdivided into a total of 774 local government areas (LGAs), and each LGA is divided into wards (the lowest administrative unit) [13]. The sample for the 2018 NDHS was selected using a two-stage stratified cluster sampling method, using 36 states and the Federal Capital Territory as urban and rural strata. In the first stage, 1400 Enumeration Areas (EAs) were randomly selected in each sampling stratum with probability proportional to EA size, and a complete household census was conducted to identify the number of households in each selected EA. In the second stage, an equal-probability systematic random sampling technique was applied to select a fixed number of 30 households per EAs. Out of 42,121 eligible women aged 15–49 years, 41,821 responded to the survey, a response rate of 99.3% [13].

The 2018 NDHS recorded information on maternal and child anthropometry, including height and weight measurements for children aged 0–59 months. To improve the quality of anthropometric measurements, the 2018 NDHS implemented data quality assurance procedures, including a subsequent day re-measurement of the height and weight of 10% of a random sample of the children. This study included 12,627 under-5 children in 1377 clusters with valid geographic coordinates and anthropometric data. The detailed methodology for the 2018 NDHS survey has been reported elsewhere [13].

The 2018 NDHS collected geographic coordinates using Global Positioning System (GPS) receivers for each cluster. To keep the confidentiality of respondents in these clusters, GPS coordinates were displaced (geo-masking) by 10 km for rural clusters and 2 km for urban clusters [27]. The NDHS also has information on climatic and demographic factors obtained from publicly available remote sensing raster and vector data sources [27]. During extraction, circular buffers of 2 km for urban points and 10 km for rural points were considered to compensate for displacements in the GPS coordinates of each cluster and differences in the pixel size of data sources [27]. Detailed procedures for extracting DHS geo-covariates were published in the DHS manual [27].

### 2.2. Outcome Variable

The study’s outcome variable was stunting, which was measured using height-for-age z-scores (HAZ) according to the WHO Child Growth Standards [28]. Height or length was measured using a Shorr measuring Board^®^ (Weigh and Measure, LLC, Olney, MD, USA) [13,29]. Children were considered stunted if their HAZ score was less than −2.0 standard deviations from the median of the WHO reference population for age. The stunting status was dichotomised as “stunted” (1) or “not stunted” (0), in line with the 2018 NDHS report [13] and previously published studies [22,30]. 

### 2.3. Explanatory Variables

Using a conceptual framework adapted from the UNICEF [31] and WHO [32] frameworks for undernutrition [32] and used in past studies from LMICs [22,30], we broadly grouped explanatory variables into proximal and contextual factors (Figure 1) [31,32]. For this study, child morbidity (including diarrhoea and acute respiratory infection (ARI)) was included as an immediate proximal factor, given the immediate pathophysiological relationships with stunting, where infection can lead to poor nutrient intake, absorption, or utilisation (Table 1) [31,32].

As shown in Table 1, proximal contextual factors included maternal factors (i.e., maternal education, nutritional status, employment status, and maternal age), household factors (i.e., household wealth, household toilet system, source of drinking water, and type of cooking fuel), health service factors (i.e., frequency of antenatal care (ANC) visits and place of birth), and child factors (i.e., birth size, child age, and birth order of the child). Environmental contextual factors included exposure to media, while climatic factors were daytime land surface temperature (DLST), aridity, rainfall, and urbanization [33,34]. We selected climatic factors based on recent studies which demonstrated increasing evidence of the impact of poverty and crop production on stunting. Climate-related data were obtained from publicly available remote sources [27].

### 2.4. Analytical Strategy

The overall analytical framework and strategy for the describing the prevalence of childhood stunting at the second administrative level while examining relevant proximal and contextual factors associated with childhood stunting have been described in detail elsewhere [30].

In summary, frequencies and percentages of proximal and contextual factors were calculated to describe the study participants. The prevalence of stunting was calculated to understand the magnitude of childhood stunting in relation to the proximal and contextual factors. To adjust for sampling weights, clustering, and stratification, all descriptive analyses, including frequencies, percentages, and prevalence, were performed using the “svydesign” function from the “survey” package in R (R Core Team, Austria) [35].

We used a five-staged analytical strategy to implement the Bayesian geostatistical modelling, as reported in detail elsewhere [30]. In stage one, data on environmental and climatic factors were linked with household survey data using a ‘cluster id’ as a unique identifier. Both the geo-covariates and the household survey data have their associated cluster-level geocoordinates. In stage two, model selection was conducted using the Watanabe–Akaike information criterion (WAIC) and deviance information criterion (DIC). We excluded maternal employment, maternal age, delivery assistance, birth order, vaccination status, birth interval media exposure (radio, television, and magazine), and climatic factors (proximity to water bodies and enhanced vegetation index) from the final model.

In stage three, by assuming childhood stunting as a continuous process in space (continuity only refers to the space domain, not the measurement scale), we implemented the stochastic partial differential equation approach (SPDE) using the R-INLA package to fit a spatial model and predict childhood stunting at unsampled locations. The spatial autocorrelation was adjusted using the Matérn covariance matrix by assuming a stationary and isotropic structure (where the correlation between two observation points only depends on their distance) [36]. As geostatistical data (point-referenced data) do not have explicit neighbouring between points, an artificial set of vertices called a mesh was created to represent the neighbouring structure (Appendix A). The observed locations (NDHS clusters) and mesh vertices (that were weighted based on their distance from the observed locations) were mapped using a projector matrix ‘A’ [36].

In stage four, the study used Bayesian geostatistical models to analyse the relationship between stunting and proximal and contextual factors. Bayesian inference was chosen for its ability to consider prior information, make predictions for unsampled locations, provide credible intervals (CrIs), and handle complex models such as spatial and spatiotemporal models [36]. In the prediction model of the prevalence of stunting in high-resolution grids [30,37], the model used proximal and contextual factors with available raster surfaces, including maternal education, ANC visits, place of birth, type of toilet system, source of drinking water, aridity, number of wet days, DLST, and altitude. In stage five, model validation was conducted by grouping the total dataset into a “training set” (75% of the sample) and a “test set” (the remaining 25% of clusters) based on previously published studies [30,38,39]. Appendix A is presented to show the correlation between the observed and predicted values as model validation. ORs with corresponding 95% CrIs were calculated and reported as the measure of association between proximal and contextual factors and stunting.

## 3. Results

### 3.1. Prevalence of Childhood Stunting

The overall national prevalence of childhood stunting was 41.5% (95% CrI: 26.4%, 55.7%). The highest prevalence of stunting reported was in Biriniwa in Jigawa State, Northern Nigeria (*p* = 66.4%; 95% CrI: 26.3%, 91.7%), followed by Yusufari in Yobe State (*p* = 66.3%; 95% CrI: 24.4%, 92.1%), while the lowest prevalence was observed in Shomolu in Lagos State Southern Nigeria (*p* = 2.0%; 95% CrI: 0.5%, 7.6%) (Table 1, Figure 2 and Figure 3). First and second administrative level prevalence estimates are presented in Appendix A.

Children who were perceived by their mothers to be very small at birth had a higher prevalence of stunting, while those who were very large at birth had a lower prevalence of stunting (52.4% vs. 28.2%). Children who experienced three episodes of diarrhoea before the survey had a higher prevalence of stunting compared to those who did not experience episodes of diarrhoea (50.0% vs. 34.6%) (Table 2).

### 3.2. Determinants of Childhood Stunting

The magnitude of associations between proximal and contextual factors and stunting was similar for both spatial and non-spatial models. Children who were perceived to be small at the time of birth were more likely to be stunted compared to those who were perceived to be very large at birth (aOR = 1.64; 95% CrI: 1.35, 1.99 for small size and aOR = 2.39; 95% CrI: 1.80, 3.17 for very small size). Children who experienced three episodes of diarrhoea in the two weeks before the survey were more likely to be stunted compared to those who did not experience diarrhoea (aOR = 1.35; 95% CrI: 1.18, 1.54) (Table 3).

Children whose mothers were overweight or obese were less likely to be stunted compared to mothers with normal weight (aOR = 0.79; 95% CrI: 0.69, 0.89 for overweight and aOR = 0.63; 95% CrI: 0.52, 0.76 for obese). Children whose mothers attained secondary or higher schooling were less likely to be stunted compared to those whose mothers did not attain any schooling (aOR = 0.69; 95% CrI: 0.60, 0.80), while children who resided in rich households were less likely to be stunted compared to those who resided in poor households (aOR = 0.75; 95% CrI: 0.64, 0.8) (Table 3).

Children who resided in households with a clean type of cooking fuel were less likely to be stunted compared to those resided in household with no clean type of cooking fuel (aOR = 0.79; 95% CrI: 0.67, 0.92). Residing in a medium rainfall geographic area (142–1199 mm rainfall) was negatively associated with childhood stunting compared to a low rainfall geographic area (aOR = 0.78; 95% CrI: 0.64, 0.96), while children who resided in urban centres (cities) were less likely to be stunted compared to those who resided in rural villages (aOR = 0.69; 95% CrI: 0.59, 0.82). Children from the North-eastern Nigeria region were more likely to be stunted compared to those who resided in the North-central region (aOR = 1.81; 95% CrI: 1.26, 2.57) (Table 3).

## 4. Discussion

This study showed striking sub-national and regional variations in the prevalence of stunting in Nigeria. Biriniwa in Jigawa State in Northern Nigeria had the highest prevalence of stunting, while Shomolu in Lagos State in Southern Nigeria had the lowest prevalence of stunting. Factors that were positively associated with stunting included perceived birth size (small at birth) and three episodes of diarrhoea in the two weeks before the survey. Children whose mothers received formal education and/or were overweight or obese were less likely to be stunted compared to compared to those who were not.

The availability of appropriate regional estimates of malnutrition is essential in order to provide the appropriate health services to the relevant population [25]. Consistent with past studies conducted in Ethiopia [30] and Rwanda [40], the current study showed wide geospatial variations of childhood stunting in Nigeria. Our findings demonstrated a high prevalence of childhood stunting in Northern Nigeria. This high burden of stunting is multifactorial, and could be attributed to the complex interplay of low education and literacy rate, poverty, inadequate health service utilisations, food insecurity, and activities of the Boko Haram (terrorist) Organisation in Northern Nigeria [13]. National nutrition programmes in Nigeria must not only be holistic but should also consider the specific geopolitical locations with the highest burden of malnutrition.

Diarrhoeal disease is the second leading cause of deaths of children aged under 5 globally (attributable to 525,000 deaths yearly), and most of these deaths are related to frequent episodes of diarrhoea each year [41]. Our study showed that children who reported three diarrhoeal episodes in the two weeks before the survey were more likely to be stunted compared to their counterparts. This finding is similar to a prior Nigerian study, which showed a positive association between diarrhoeal episodes and stunting [22]. Diarrhoeal disease leads to childhood malnutrition in the following pathways: (i) through reduced appetite and increased catabolism; (ii) impaired intestinal absorption due to the altered barrier and mucosal inflammation [42]; and (iii) displacements of essential nutrients (e.g., minerals and vitamins) for the immune response [22,42]. Our findings suggest the need to promote national policies and investments that support case management of common childhood illnesses (including diarrhoea) and their complications, as well as to increase access to safe drinking water and sanitation in Nigeria.

Maternal nutritional intake during pregnancy negatively affects the division and replication of cells in the embryo and the development of the foetus in later stages of pregnancy, with subsequent impacts on the birth size, nutritional status, and development of the baby [43]. Consistent with this evidence and previously published studies conducted in Ethiopia [30] and Tanzania [44], children whose mothers perceived them as small or very small in size at the time of delivery were more likely to be stunted compared to their counterparts. Poor fat and protein absorption, difficulties in initiating breastfeeding, infections, and hospitalisations [45] are associated with small-birth-size children. This association possibly explains the relationship between small-birth-size children and stunting. Our findings suggest the need for interventions to improve the nutritional status of women before and during pregnancy to eliminate the intergenerational influence of malnutrition.

Past studies have showed the protective effect of living in improved socioeconomic neighbourhoods on the nutritional status of children [1,46]. Studies from Ethiopia [30] and Tanzania [44] showed that higher socioeconomic households were associated with a lower risk of childhood stunting. There are several reasons for the protective effect of improved socioeconomic status on childhood malnutrition, including: (i) it increases the household decision-making power of women (including a decision on food purchases) [47]; (ii) it has positive impacts on household income and food purchasing power [48]; and (iii) it improves the healthcare-seeking behaviours of parents for timely vaccination and treatment of the sick child [49]. Improvement of programmes that target disadvantaged socioeconomic neighbours in Northern Nigeria and urban slums in Southern Nigeria would help to improve the nutritional status of Nigerian children.

This study showed that children who resided in geographic locations with medium rainfall were less likely to be stunted, consistent with findings from Ethiopia [50]. Extreme rainfall and climatic variability have negative implications on crop production, household food security, and malnutrition. This is particularly common in sub-Saharan African countries (including Nigeria), where crop production is predominantly based on rain-fed agriculture [51,52]. The effect of climate variability on food production is not only prominent in sub-Saharan African countries, but the utilisation of adaptation and resilient mechanisms (such as agricultural irrigation) is inadequate [51,52]. Our findings suggest the need for adaptation measures that sustainably increase water availability (e.g., irrigation) for crop production and diversification of agriculture production (e.g., crops, livestock, and fisheries) to subsequently improve the nutritional status of Nigerian children.

Research on urban–rural disparities of malnutrition showed that urban children (apart from those living in urban slums) generally have improved nutritional status compared to rural children [53,54]. These studies also showed that urban children have better access to food, good housing, employment opportunities for parents, access to health services, and basic amenities (e.g., electricity, water, and sanitation) [53]. Findings from the current study showed that children who resided in urban cities (apart from those residing in the slums) were less likely to be stunted compared to their counterparts. The finding is similar to evidence from a Tanzanian study [44]. However, a study conducted in Zambia [55] showed a positive association between urban children and stunting. Although the present study showed that urban children had a lower risk of stunting, there is also a need for further research on urban–rural disparities and its determinants to examine the proximal and contextual factors separately.

### Strengths and Limitations of the Study

The study has the following limitations. First, the inference of causality between the measured exposure variables and the outcome (stunting) is challenging given the use of a cross-sectional dataset. Nevertheless, the study findings are consistent with results from other studies that used cross-sectional data or more robust study designs. Additionally, the analyses of some study factors (e.g., climatic factors) can only be performed with data obtained from observational study designs. Second, unobserved potential confounders, including food insecurity and social networks, may affect the results of our findings. Nevertheless, climatic factors (e.g., aridity, rainfall, and temperature) can be proxy indicators for agricultural production and food security [33]. Finally, non-differential misclassification bias due to the displacement of GPS coordinates is possible. However, we used a circular buffer while extracting geo-covariates from each cluster [27]. Despite these limitations, the study has strengths. The national representativeness of the DHS data increases the generalisability of the findings. The study used Bayesian inference to utilise its superior strength in spatial modelling and the production of small-surface-area estimates. Additionally, we investigated the association between environmental and climatic factors with stunting, and these results provided detailed understanding of the public health issues relating to childhood malnutrition in Nigeria.

## 5. Conclusions

The study showed wide variations in stunting prevalence across the second-level administrative regions in Nigeria, where Biriniwa in Jigawa State in Northern Nigeria had the highest prevalence of stunting, and Shomolu in Lagos State in Southern Nigeria reported the lowest prevalence. Children who were perceived to be small, those who experienced diarrhoea, and those living in disadvantaged areas in the north or urban slums in the south were more at risk of being stunted. The results highlight the importance of investing and implementing targeted health and socioeconomic policies to reduce childhood illnesses (such as diarrhoea) and access to safe water and sanitation in Nigeria. Targeting disadvantaged neighborhoods in Northern Nigeria and urban slums in Southern Nigeria through programmes can improve child nutrition. Further research is also needed to investigate the urban–rural disparities in stunting and its determinants by urban–rural status specifically.

## Figures and Tables

**Figure 1 ijerph-20-03250-f001:**
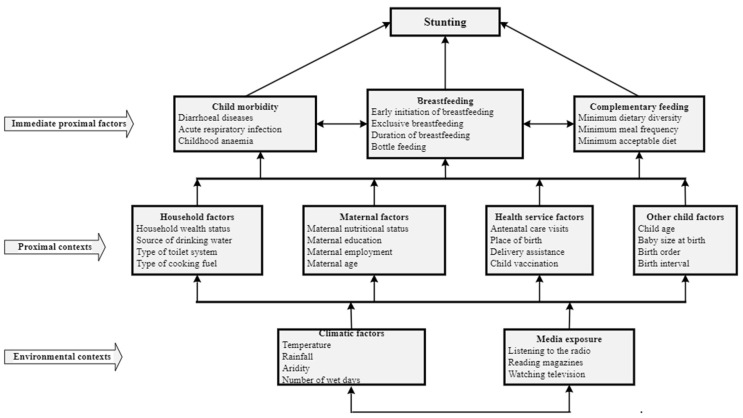
Conceptual framework for proximal and contextual determinants of stunting among children under five years of age (adapted from UNICEF 2013 [26] and WHO 2017 [27]).

**Figure 2 ijerph-20-03250-f002:**
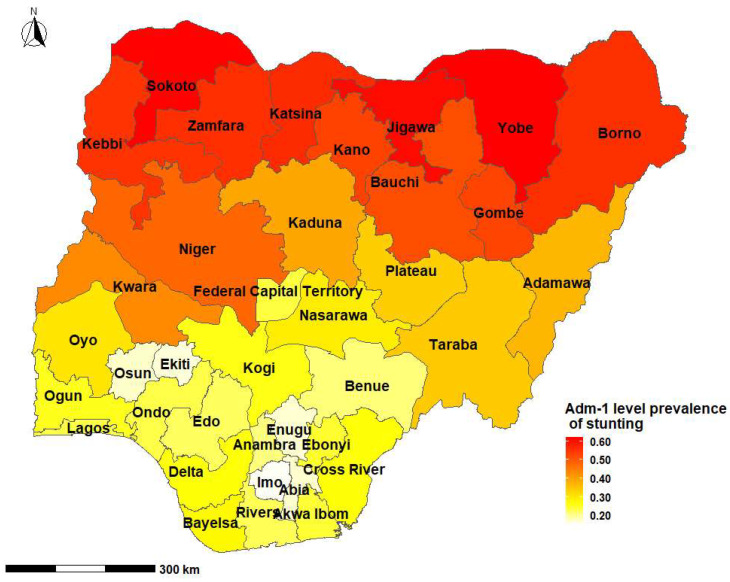
Prevalence of childhood stunting in Nigeria by states and the Federal Capital Territory, 2018.

**Figure 3 ijerph-20-03250-f003:**
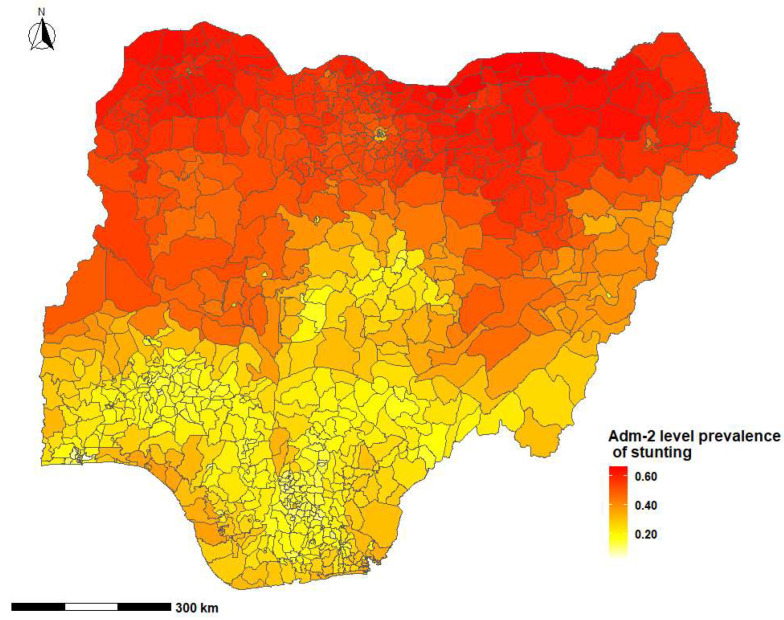
Prevalence of childhood stunting across the second administrative level in Nigeria by Local Government Councils, 2018.

**Table 1 ijerph-20-03250-t001:** Definition of proximal and contextual factors associated with childhood stunting among children under five years.

Explanatory Variable	Definitions	Reference Category	Data Sources
**Immediate proximal factors**			
Diarrhoea	Diarrhoea was defined as a passing of abnormal stools during the two weeks preceding the survey. Diarrhoea was grouped as ‘1’ = ‘experienced diarrhoea’ or ‘2’ = ‘did not experience diarrhoea’	Did not have diarrhoea’	Self-reports
Acute respiratory infection (ARI)	ARI was defined as symptoms of cough and shortness of breath during the two weeks preceding the survey. ARI was grouped as ‘1’ = ‘experienced ARI’ or ‘2’ = ‘did not experience ARI’	No	Self-reports
**Proximal contextual factors**			
Maternal nutritional status	Maternal nutrition was grouped as ‘1’ = ‘underweight (BMI < 18.5 kg/m^2^)’, ‘2’ = ‘normal weight (BMI ≥ 18.5 kg/m^2^ and BMI ≤ 24.9 kg/m^2^)’, or ‘3’ = ‘overweight or obese (BMI ≥ 25.0 kg/m^2^)’	Normal weight	Anthropometric measurements
Mothers’ and fathers’ educational status	Mothers’ and fathers’ educational status were grouped as ‘1’ = ‘no schooling’, ‘2’ = ‘primary education’, or ‘3’ = ‘secondary education or higher’	No schooling	Self-reports
Mothers’ and fathers’ employment	Mothers’ and fathers’ employment were grouped as ‘1’ = ‘no employment’, ‘2’ = ‘formal employment’, or ‘3’ = ‘informal employment’	No employment	Self-reports
Maternal age	Maternal age was grouped as ‘1’ = ‘15–24 years’, ‘2’ = ‘25–34 years’, or ‘3’ = ‘35–49 years’.	15–24 years	Self-reports
Household wealth index	Household wealth index was grouped as ‘1’ = ‘poor’, ‘2’ = ‘middle’, or ‘3’ = ‘rich’	Poor	Self-reports
Source of drinking water and type of toilet facility	The source of drinking water and type of toilet facility were grouped as ‘1’ = ‘improved’ or ‘2’ = ‘not improved’	Not improved	Self-reports
Type of cooking fuel	Type of cooking fuel was grouped as ‘1’ = ‘cleaned’ or ‘2’ = ‘not cleaned’	Not cleaned	Self-reports
Frequency of ANC visits	Frequency of ANC visits was grouped as ‘1’ = ‘none’, ‘2’ = ‘1–4 visits’, or ‘3’ = ‘four and above visits’	None	Self-reports
Place of birth	Place of birth of was grouped as ‘home ‘or ‘health facility birth’	Home	Self-reports
Perceived birth size	Perceived birth size was grouped as ‘1’ = ‘larger than average’, ‘2’ = ‘average’, or ‘3’ = ‘smaller than average’, and birth order (categorized as ‘1’, ‘2–4’, or ‘5 and above’).	Larger than average	Self-reports
**Environmental contextual factors**			
Media exposure factors	Media exposure factors included listening to radio, reading magazine, and watching television. Media exposure factors were grouped as ‘1’ = ‘Yes’ or ‘2’ = ‘No’	No	Self-reports
Mean annual daytime land surface temperature (DLST)	DLST was defined daytime temperature of the land surface (skin temperature), detected by satellites by looking through the atmosphere to the ground [27]. The DLST was grouped as ‘1’ = ‘<30.0 °C’, ‘2’ = ‘30–34.9 °C’, or ‘3’ = ‘+35.0 °C’.	<30.0 °C	Spatial resolutions (6 km by 6 km) obtained from MODIS
Mean annual rainfall	Mean annual rainfall was grouped as ‘1’ = ‘low’ (less than 142 mm), ‘2’ = ‘medium’ (142–1200 mm), or ‘3’ = ‘high’ (more than 1200 mm).	Low	Spatial resolutions (5 km by 5 km) obtained from CHIRPS data 2.0
Aridity	Aridity was defined as the mean annual precipitation divided by mean annual potential evapotranspiration ranged from 0 (most arid) to 300 (most wet) [27]. The aridity index was grouped as ‘1’ = ‘arid’, aridity index of less than 17.5, ‘2’ = ‘semi-arid’, aridity index from 17.5 up to 32.5, or ‘3’ = ‘wet’, aridity index above 32.5	Wet	Spatial resolution for covariates (55 km by 55 km) obtained from CRU datasets
Urbanisation	Urbanisation was defined based on the urban–rural settlement classification. Cities were grouped as ‘1’ = ‘urban centres’, towns and suburbs were grouped as ‘2’ = ‘urban clusters’, and rural villages were grouped as ‘3’ = ‘rural villages’.	Low	Spatial resolution for covariates (55 km by 55 km) obtained from CRU datasets

NB: The definition, classification, and data sources of the explanatory variables were based on a previously published study [30].

**Table 2 ijerph-20-03250-t002:** Prevalence of stunting over proximal and contextual determinants among children under five years of age in Nigeria, 2018 NDHS (n = 12,627).

Variables	n (%)	Childhood Stunting	
		Yes, n (%)	No, n (%)
**Immediate proximal factors**			
Diarrhoeal diseases			
No	10,242 (87.4)	4333 (36.7)	7479 (63.3)
Yes	1473 (12.6)	270 (39.1)	420 (60.9)
Acute respiratory infection			
No	12,112 (94.5)	3471 (34.6)	6577 (65.4)
Yes	702 (5.5)	726 (50.0)	726 (50.0)
**Proximal contextual factors**			
Maternal nutritional status			
Underweight	1080 (9.4)	534 (49.9)	535 (50.1)
Normal	7063 (61.4)	2841 (40.5)	4175 (59.5)
Overweight	2246 (19.5)	573 (25.7)	1655 (74.3)
Obese	1112 (9.7)	210 (19.1)	983 (80.9)
Maternal age			
15–24 years	2488 (11.2)	987 (40.6)	1443 (59.4)
25–34 years	6082 (51.9)	2082 (34.9)	3883 (65.1)
35–49 years	3147 (26.9)	1131 (36.4)	1976 (63.6)
Maternal educational status			
No schooling	4546 (38.8)	2403 (54.1)	2043 (45.9)
Primary education	1864 (15.9)	698 (37.9)	1143 (62.1)
Secondary or higher education	5308 (45.3)	1099 (21.1)	4116 (78.9)
Household wealth status			
Poor	4433 (37.8)	2277 (52.3)	2079 (47.7)
Middle	2427 (20.7)	893 (37.5)	1491 (62.5)
Rich	4858 (41.5)	1029 (21.6)	3732 (78.4)
Antenatal care visits			
None	1621 (20.9)	773 (48.7)	815 (51.3)
1–3 visits	1238 (16.0)	502 (41.5)	707 (58.5)
4+ visits	4889 (63.1)	1377 (28.6)	3438 (71.4)
Place of birth			
Home	6384 (54.5)	2967 (47.3)	3300 (52.6)
Health facility	5334 (45.5)	1233 (23.5)	4003 (76.4)
Mother’s perceived baby size at birth			
Very large	1061 (9.1)	296 (28.2)	752 (71.8)
Larger than average	2845 (24.3)	1042 (37.3)	1750 (62.7)
Average	6367 (54.3)	2236 (35.8)	4017 (64.2)
Smaller than average	1156 (9.9)	493 (43.7)	635 (56.3)
Very small	289 (2.5)	134 (52.4)	148 (52.4)
Cooking fuel type			
Not clean	8742 (74.6)	3693 (43.0)	4895 (57.0)
Clean	2976 (25.4)	507 (17.4)	2408 (82.6)
Source of drinking water			
Not protected	6673 (52.1)	2606 (40.3)	3854 (59.7)
Protected	6141 (47.9)	1998 (33.1)	4045 (66.9)
Toilet system			
Not improved	5272 (45.0)	2281 (44.0)	2906 (56.0)
Improved	6446 (55.0)	1919 (30.4)	4396 (69.6)
**Environmental contextual factors**			
Daytime land surface temperature			
<30 °C	3471 (27.1)	761 (22.5)	2626 (77.5)
30–34.99 °C	6366 (49.7)	2333 (37.4)	3902 (62.6)
+ 35 °C	2676 (23.2)	1510 (52.4)	1371 (47.6)
Annual average rainfall (in mm)			
< 141 mm	7823 (61.1)	3554 (46.5)	4082 (53.5)
142–1199 mm	4518 (35.2)	942 (21.4)	3459 (78.6)
>= 1200 mm	474 (3.7)	107 (23.1)	358 (76.9)
Aridity			
Wet	1920 (14.2)	360 (20.3)	1413 (79.7)
Semi-arid	4267 (33.3)	923 (22.1)	3247 (77.9)
Arid	6727 (52.5)	3320 (50.6)	3238 (49.4)
Urbanization			
Low	8163 (63.70)	3480 (43.6)	4495 (56.4)
Medium	605 (4.7)	209 (35.2)	385 (64.8)
High	4047 (31.6)	914 (23.3)	3018 (76.7)
Regions			
North-central	1788 (14.0)	510 (28.8)	1258 (71.2)
North-east	2051 (16.0)	964 (48.9)	1007 (51.1)
North-west	3629 (28.3)	2016 (56.8)	1532 (43.2)
South-east	1676 (13.1)	298 (18.1)	1349 (81.9)
South-south	1349 (10.5)	255 (19.6)	1043 (80.4)
South-west	2320 (18.1)	561 (24.7)	1710 (75.3)

**Table 3 ijerph-20-03250-t003:** Proximal and contextual determinants of childhood stunting among children under five years in Nigeria, 2018 NDHS (n = 12,627).

Variables	Childhood Stunting	
	Non-Spatial Model OR (95% Crl)	Geospatial Model OR (95% Crl)
**Immediate proximal factors**		
Diarrhoeal diseases		
No	1.00	1.00
Yes	1.37 (1.20, 1.56) *	1.35 (1.18, 1.54) *
Acute respiratory infection		
No	1.00	1.00
Yes	0.99 (0.83, 1.19)	0.99 (0.83, 1.19)
**Proximal contextual factors**		
Maternal nutritional status		
Normal	1.00	1.00
Underweight	1.15 (0.99, 1.33)	1.14 (0.99, 1.32)
Overweight	0.79 (0.69, 0.89) *	0.79 (0.69, 0.89) *
Obese	0.62 (0.51, 0.75) *	0.63 (0.52, 0.76) *
Maternal educational status		
No schooling	1.00	1.00
Primary education	0.94 (0.82, 1.08)	0.92 (0.80, 1.06)
Secondary or higher education	0.70 (0.60, 0.81) *	0.69 (0.60, 0.80) *
Household wealth status		
Poor	1.00	1.00
Middle	0.93 (0.82, 1.06)	0.94 (0.82, 1.07)
Rich	0.75 (0.63, 0.88) *	0.75 (0.64, 0.89) *
Antenatal care visits		
None	1.00	1.00
1–3 visits	1.09 (0.94, 1.26)	1.08 (0.94, 1.26)
4+ visits	1.04 (0.94, 1.16)	1.04 (0.94, 1.16)
Place of birth		
Home	1.00	1.00
Health facility	0.86 (0.75, 0.98)	0.86 (0.75, 0.99)
Mother’s perceived baby size at birth		
Very large	1.00	1.00
Larger than average	1.26 (1.07, 1.48) *	1.27 (1.08, 1.49) *
Average	1.32 (1.14, 1.52) *	1.32 (1.14, 1.53) *
Smaller than average	1.62 (1.34, 1.97) *	1.64 (1.35, 1.99) *
Very small	2.35 (1.77, 3.11) *	2.39 (1.80, 3.17) *
Cooking fuel type		
Not clean	1.00	1.00
Clean	0.81 (0.69, 0.95) *	0.79 (0.67, 0.92) *
Source of drinking water		
Not protected	1.00	1.00
Protected	0.97 (0.89, 1.07)	0.97 (0.88, 1.07)
Toilet system		
Not improved	1.00	1.00
Improved	1.02 (0.92, 1.14)	1.04 (0.93, 1.16)
**Environmental contextual factors**		
Daytime land surface temperature		
<30 °C	1.00	1.00
30–34.99 °C	1.08 (0.90, 1.30)	1.01 (0.84, 1.23)
+35 °C	1.22 (0.96, 1.55)	1.11 (0.86, 1.43)
Annual average rainfall (in mm)		
<141 mm	1.00	1.00
142–1199 mm	0.75 (0.63, 0.89) *	0.78 (0.64, 0.96) *
≥1200 mm	0.72 (0.51, 1.01)	0.77 (0.53, 1.12)
Aridity		
Wet	1.00	1.00
Semi-arid	1.24 (0.93, 1.66)	1.16 (0.82, 1.64)
Arid	1.60 (1.02, 2.51)	1.50 (0.89, 2.54)
Urbanization		
Rural villages	1.00	1.00
Urban clusters	0.83 (0.65, 1.07)	0.83 (0.64, 1.06)
Urban centres	0.69 (0.58, 0.81) *	0.69 (0.59, 0.82) *
Regions		
North-central	1.00	1.00
North-east	1.35 (1.06, 1.72) *	1.23 (0.87, 1.71)
North-west	1.93 (1.49, 2.51) *	1.81 (1.26, 2.57) *
South-east	0.85 (0.65, 1.12)	0.89 (0.59, 1.35)
South-south	0.81 (0.57, 1.13)	0.75 (0.49, 1.15)
South-west	1.74 (1.38, 2.19) *	1.43 (0.97, 2.06)
**Model validation**		
DIC	13,949.06	13,927.15
WAIC	13,949.13	13,926.82
Marginal likelihood	−7299.42	−7292.09
**Spatial random effects**		
Kappa	-	7.33 (2.14, 11.86)
Variance	-	0.24 (0.10, 0.39)
* Range (in km)	-	52.2 (15.5, 98.8)

* Indicates statistical significance; OR = Odds Ratio; 95% Crl = 95% Credible Interval; DIC = Deviance Information Criterion; WAIC = Watanabe–Akaike Information Criterion. * Range indicates the distance value (in the unit of the point coordinates) above which spatial dependencies become negligible.

## Data Availability

Datasets are available at https://www.dhsprogram.com/data/ (accessed on 15 November 2021).

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
