# Peer review of "Mapping Local Variations and the Determinants of Childhood Stunting in Nigeria"

_ijerph, 2023, doi:10.3390/ijerph20043250_

Round 1

Reviewer 1 Report

Dear authors, 

After a careful reading of your manuscript, I found it very interesting and with a topic that is appropriate to the journal. 

The theme is very important for the society we are all living in, because despite many strategies for development and many goals poor countries remain in the same difficult position with insufficient financial resources and implemented policies in order to redress real problems. 

A part of the authors have published previously a similar analysis in Nutrients, but applied to a different country Ethiopia. If you check for similarity, you may find that at page 3 (lines 99-142), page 4 - the figure, page 5 (lines 151-164 and 166-180), page 7 (lines 203-217) the formulations are almost copy pasted from the previous study.

It is sad because the entire endeavor is very interesting and the amount of work deposited as to finish the research was also significant. In my opinion, these parts of the paper, that explain actually the main body of the research design - the strategy and the variables - should be refined. 

Otherwise, the analysis is interesting and the dimensions taken into consideration as to better capture the determinants of childhood stunting in Nigeria are also relevant for other poor countries.

In order to better shape the phenomenon of childhood stunting and its determinants, the adequacy of the literature review can be improved. Few papers were nominated in the introductory part, and I found it as being insufficient for the topic of the paper. 

Also, conclusions are, in my opinion, too short. They could be further expanded as to summarize the main findings of the analysis. 

Finally, I appreciate that the authors have pointed out the novelty of their analysis and also the limitations of their research. 

Reviewer 2 Report

This is an interesting paper that aims to describe the prevalence of childhood stunting at the second administrative level in Nigeria, and to examine proximal and contextual factors associated with childhood stunting like child morbidity, household factors, maternal factors, health service factors and the geospatial influence. The results from this study are useful for policy decision-makers and public health experts in order to scale-up nutrition-specific interventions in this country highly affected by malnutrition and which accounts for 8% of the world's stunted children according to the evidence reported in the introduction. Although this problem has been explored previously, this report adds important information about geography-specific variations of stunting in Nigeria taking into account environmental and climatic factors, topics of vital relevance to the region's economy and to the public health policies to be adopted. Methods are clearly described. The source of data is reliable, since it is from the 2018 Nigeria Demographic and Health Survey (NDHS). The sampling technique is well described as well as the statistical analysis. Results are consistent with the study objectives. Conclusions are based on results and discussion talks about evidence that support findings and novel knowledge apported by the paper. Additionally, strengths and limitations are stated. I congratulate authors for their article.

Major comments: bearing in mind that the abstract is the way you make your article known, I suggest that it follows a structure of introduction, objectives, methods, results, conclusions that allows the reader to get an idea of the content of the article.

Reviewer 3 Report

The authors described local variations in the prevalence of childhood stunting at the second administrative level and its determinants in Nigeria after accounting for the influence of geospatial dependencies. This study used the 2018 national Nigeria Demographic and Health Survey datasets (N = 12,627).

Comments:

Fig, 1

·        Fasting is listed in the top box.  However no data is presented for fasting.

·        In presented study, child morbidity (including diarrhea and acute respiratory infection was included as an immediate proximal factor while in fig 1 they are presented as “proximal causes”.  The wording should be consistent.

Table 3

·        Proximal and contextual determinants of childhood stunting among children under-five years in Nigeria are presented. However, it is not clear to which group the given factor belongs.

·        The differences ( in the results obtained by non-spatial and geospatial analyses are not explained. Lines 361-367

Conclusions

·        Conclusions are not prioritizing the  factors that are planned for future interventions.The scope of the stunting problem in Nigeria is extremely large and the progress can be archived only by step by step approach.  However, the authors are not taking into account the complex information about the relationship found between the presence of shunting and a wide range of proximal and environmental risk factors. No attempt was undertaken to indicate the modifiable risk factors taking into country resources.It is well known that children who were small at birth, and those who resided in a disadvantaged socioeconomic neighbourhood are at higher risk of being stunted. 

·        One of the proposed directions for future research is to clarify the role of urban-rural disparities of stunting related to the socio-economic characteristics of the families living in these areas. However, it seems that such hypotheses may be analyzed based on obtained data.  

Reviewer 4 Report

Congratulations to the authors for conducting such an important study in the current scenario, which will reduce the rate of stunting. Here are some areas for improvement. 

Introduction: Provide greater depth of the previous theoretical framework that allows knowing what aspects are known prior to the research and what the knowledge gap is. 

Methods: Have the authors considered performing a multivariate analysis to determine the effect of the variables jointly?

Results: Add the P value in Table 3 to make it easier to assess which contrasts are significant and which are not.

Conclusion: To provide greater depth of conclusions based on the results obtained.

Round 2

Reviewer 1 Report

Dear Authors, 

Thank you for the revised version of your manuscript.

The only aspect that captured my attention, in this stage is that in the abstract, you don't have to specify "introduction", "objectives", results, etc. These aspects should be naturally understood from the text.

In rest, I consider that the new version of the paper in improved and ready to be published in the current form.

Thank you!

Reviewer 3 Report

I do not have any further comments. The manuscript may be accepted.

Reviewer 4 Report

The manuscript has been considerably improved